# The Role of the Myokine Irisin in the Protection and Carcinogenesis of the Gastrointestinal Tract

**DOI:** 10.3390/antiox13040413

**Published:** 2024-03-28

**Authors:** Monika Pinkas, Tomasz Brzozowski

**Affiliations:** 1Department of Physiology, Faculty of Medicine, Jagiellonian University Medical College, 16 Grzegorzecka Street, 31-531 Cracow, Poland; monika.pinkas@doctoral.uj.edu.pl; 2Doctoral School of Medical and Health Sciences, Jagiellonian University Medical College, 31-008 Cracow, Poland

**Keywords:** irisin, myokines, exercise, physical activity, inflammation, gastrointestinal diseases, cancer

## Abstract

Recently discovered irisin, a member of the myokines family, is a potential mediator of exercise-induced energy metabolism and a factor promoting browning of the white adipose tissue. Recent evidence indicates that this myokine, released from contracting muscles, can mediate the beneficial effects of exercise on health. Irisin may be a potential therapeutic agent against obesity and has been shown to play an important role in the protection of various cells, tissues, and organs due to its anti-inflammatory, antioxidative, and anti-cancer properties. Our aim was to review the recent experimental and clinical studies on irisin and its expression, release into the bloodstream, tissue targets, and potential contribution to the protective effects of exercise in the gastrointestinal tract. Particular emphasis was placed on inflammatory bowel disease, intestinal ischemia/reperfusion injury, periodontitis, and other digestive tract disorders, including carcinogenesis. Overall, irisin holds significant potential as a novel target molecule, offering a safe and therapeutic approach to treating various gastrointestinal diseases.

## 1. Introduction

For many years, it has been known that physical activity plays a significant role in maintaining good health and general well-being. It has been documented that regular physical activity can act prophylactically against a variety of diseases, such as cardiovascular and metabolic disorders, neurological diseases, osteoporosis, and cancer. Despite that, the underlying molecular and cellular mechanisms involved in the promotion of the countless health benefits of exercise are still far from being fully understood [1,2]. Over the years of research, it has been established that contracting muscle fibers exert autocrine, paracrine, and endocrine functions [2]. Experimental studies have proven that skeletal muscle can secrete soluble factors called myokines, such as myostatin, β-aminoisobutyric acid, interleukin-15 (IL-15), meteorin-like, and myonectin, that can positively or negatively influence the metabolism of other tissues [3]. These advances have shed a new light on our understanding of the mechanisms by which physical activity benefits health [1,2,3].

## 2. Protective Role of Exercise in the Gastrointestinal Tract

According to Global Cancer Statistics 2020, stomach cancer ranks fifth most common and fourth most lethal, while esophageal cancer ranks seventh in terms of incidence and sixth in mortality globally [4]. Moreover, a recent large-scale multinational study [5] revealed that more than 40% of the world population suffers from functional gastrointestinal (GI) disorders, such as irritable bowel syndrome (IBS). Functional GI disorders are manifested by motility disorders, visceral hypersensitivity, altered mucosal and immune function, pathological gut microbiota abundance, and negative influences on the central nervous system. The prevalence of these functional disorders lowers the quality of patients’ lives, affects cost-effectiveness, and requires more frequent physician consultations [5].

Exercise can exert a protective role against upper and lower GI tract disorders such as reflux esophagitis, peptic ulcers, colon cancer, cholelithiasis, constipation, and inflammatory bowel disease, leading to the attenuation of the symptoms [6]. Moreover, physical activity has been shown to positively influence gut microbiome biodiversity and GI barrier functioning [7]. Furthermore, evidence from a meta-analysis by Xie et al. indicates that moderate to high-intensity exercise can significantly lower the overall risk of cancer development in the digestive system [8]. The protective role of exercise in health maintenance depends on the reduction in body fat and the subsequent beneficial metabolic effects. While single bouts of exercise trigger systemic inflammation followed by immunosuppression, regular voluntary exercise has been shown to promote anti-inflammatory actions [6].

## 3. A Novel Myokine—Irisin

In 2012, Spiegelman et al. described a new myokine and named it “irisin” after the Greek messenger goddess Iris [9,10]. As demonstrated in their study [9], irisin is a proteolytically cleaved and glycosylated fragment of the fibronectin type III domain containing 5 (FNDC5), a type I transmembrane glycoprotein identified in 2002 [9,11]. Expression of FNDC5 is regulated by the transcriptional co-activator peroxisome proliferator-activated receptor-γ (PPAR-γ) co-activator-1α (PGC-1α), which mediates many biological programs related to energy metabolism associated with the beneficial effects of exercise. They demonstrated that irisin is secreted from murine muscle into circulation in response to exercise and can induce changes in the subcutaneous adipose tissue by increasing the expression of mitochondrial uncoupling protein 1 (UCP1) and activating its thermogenic function [9,10]. Moreover, irisin induces browning of the white adipose tissue (WAT) both in vitro and in vivo [9]. Human and mouse irisin are 100% identical, which is why Boström et al. [9] have suggested that its function is highly conserved across species. They examined muscle biopsies and plasma samples obtained from eight human subjects before and after 10 weeks of endurance exercise [9]. The analysis showed an upregulation of FNDC5 gene expression in beige fat and muscle tissues as well as an increase in circulating irisin levels. The obtained results were comparable to those observed in mice. In addition, they found that even moderately increased irisin secretion in mice was accompanied by a reduction in body weight and an improvement in diet-induced insulin resistance. They concluded that irisin could be responsible for at least some of the beneficial effects of physical activity and may hold significant therapeutic potential for diabetes and obesity, as well as many other diseases [9]. In this review, we will focus on the potential protective role of this novel peptide in GI tract disorders.

## 4. Detection of Irisin

The detection of irisin in human plasma has become a subject of discussion due to the prominent cross-reactivity with non-specific proteins of commercial antibodies and enzyme-linked immunosorbent assay (ELISA) kits, including the antibody used to detect this novel myokine by Boström et al. [12,13]. Moreover, different commercial ELISA kits have been reported to detect irisin levels in a range from picograms to micrograms per milliliter of serum or plasma [14,15]. In response, Jedrychowski et al. performed a precise quantitative assay of human irisin in plasma using mass spectrometry, providing substantial progress in the detection of this peptide [16]. Mass spectrometry is considered the gold standard, but only a few studies have employed this method [14]. ELISA is currently the most common method for detecting irisin in humans [15]. Some ELISA kits (e.g., kits by Aviscera and Phoenix Pharmaceuticals) have been validated by Western blot and shown to correctly detect spiked irisin under physiological conditions [17]. However, ELISA measurements are still considered to only reflect the trends of the changes in irisin’s concentrations [15]. Therefore, using different kits in the same study and comparing measurements from different companies’ ELISA irisin kits should be avoided [18]. It is also crucial to use kits validated against Western blots and mass spectrometry [13].

Currently, the physiological role and functions of irisin, as well as its potential targets, require further investigation [19,20]. One of the recent advances in research on irisin is the identification of the αVβ5 integrin as an irisin receptor in osteocytes and adipose tissues [21]. Mu et al. have shown that extracellular heat shock protein 90α acts as a co-factor for irisin’s binding with this integrin [22]. Moreover, irisin has been shown to bind to the αVβ5 receptor on gut epithelial cells both in vitro and in vivo, stressing its potential role in the digestive system [23]. Presumably, this peptide could exert beneficial effects by interacting with this integrin on various tissues, including the tissues of the GI tract [23,24,25].

## 5. Irisin’s Secretion and Mechanisms of Action

Irisin is primarily produced by skeletal muscles; however, it has also been reported that it can be secreted by adipose tissue and in smaller amounts by the testes, liver, pancreas, brain, spleen, heart, and stomach [26]. It has been shown that irisin is expressed in response to exercise as well as cold-induced muscle shivering [27,28]. However, the most effective training protocol for increasing irisin levels has not yet been established. Moreover, some individuals may show no irisin response to exercise, regardless of its type, intensity, and duration [29]. Factors like age, sex, fitness level, and body mass index may influence the effectiveness of different types of training in raising irisin levels [30]. A meta-analysis by Kazeminasab et al. demonstrated that acute aerobic training and chronic resistance training may evoke a most pronounced effect on circulating irisin in healthy subjects [31]. Moreover, results of the recent meta-analysis by Torabi et al. indicate that exercise can positively impact irisin levels in obese and overweight individuals, regardless of its type, and that high-intensity interval training might be especially effective for males [30]. Improvement and standardization of detection methods for irisin may help establish the most efficient exercise protocols in the future.

Exercise induces the expression of PGC-1α in skeletal muscle fibers, which leads to the production of FNDC5, a precursor of irisin [32]. PGC-1α interacts with several transcription factors to regulate FNDC5 synthesis, such as transcription factors estrogen-related receptor α and cyclic adenosine monophosphate (AMP) response element-binding protein, together with nuclear hormone receptors such as glucocorticoid receptor, constitutive androstane receptor, and farnesoid X receptor that increase FNDC5 mRNA expression [33]. It had been reported that, unlike in other species, the start codon of the human FNDC5 gene is mutated to the non-canonical ATA, potentially affecting the efficiency of its translation [34]. However, recently, Witmer et al. published new evidence for an upstream ATG start codon of the FNDC5 gene in humans, potentially resolving the controversy [35]. The mechanisms associated with FNDC5 cleavage into irisin, and its release are still not fully understood; however, it is thought that disintegrin and metalloproteinase (ADAM) family members are involved in the process [33]. FNDC5 consists of an N-terminal signal peptide, a fibronectin III domain, a linker domain, a hydrophobic transmembrane domain, and a cytoplasmic C-terminal domain. The signal peptide and N-glycosylation are removed, and FNDC5 is translocated to the membrane. FNDC5 is cleaved by an undetermined protease between the fibronectin III domain and the linker domain into irisin. It is speculated that angiotensin II may promote FNDC5 cleavage by upregulating ADAM-10. Irisin with a partial C-terminal domain or even uncleaved FNDC5 can be further released into the bloodstream [36].

Irisin suppresses the production of pro-inflammatory cytokines in immune and fat cells, counteracting obesity-related inflammation [37]. It exerts many functions through multiple signaling pathways. One of the main pathways regulated by irisin is the mitogen-activated protein kinase pathway (MAPK). MAPK is involved in gene expression, mitosis, metabolism, motility, cell survival, apoptosis, and differentiation. It plays an important role in the browning of white adipocytes and the proliferation and differentiation of osteoblasts. Irisin has also been shown to regulate phosphatidylinositol 3-kinase (PI3K)/protein kinase B (AKT) and signal transducer and activator of transcription 3 (STAT3)/Snail pathways that are involved in cancer cell growth, proliferation, survival, and migration [38].

Zhang et al. [39] demonstrated that irisin can promote an increase in adipocyte thermogenesis in human adipocytes through extracellular signal-regulated kinase (ERK) and p38 MAPK signaling, ultimately leading to weight loss and an improvement in insulin sensitivity. A recent report by Luo et al. [40] revealed that irisin can activate the AMP-activated protein kinase (AMPK) α1 pathway to promote the browning of WAT by upregulating the mRNA and protein expression of UCP1, PGC-1α, and the positive regulatory domain containing 16 [40]. Additionally, irisin may regulate the production of IL-6, IL-1β, monocyte chemotactic protein 1, and tumor necrosis factor-α (TNF-α) in adipocytes through the regulation of downstream signaling of Toll-like receptor 4/myeloid differentiation primary response 88 (TLR4/MyD88) [37].

It has been shown that FNDC5 mRNA expression and irisin secretion are increased in the early stages of adipogenesis. Irisin can regulate adipogenesis through the Wnt pathway by blocking the induction of PPAR-γ and CCAAT/enhancer-binding protein α and consequently inhibiting adipocyte differentiation [41]. Another role of this novel myokine in the differentiation of adipocytes has been demonstrated by Shaw et al. [42]. Irisin regulated the release of cytokines from differentiating adipocytes at least partially through upregulation of the nuclear factor-κ light chain enhancer of activated B cells (NFκB) pathway. The authors suggested that irisin might play an important role in the adhesion of endothelial cells, leading to an improvement in tissue vascularization [42]. Interestingly, in cells exposed to inflammatory signals, irisin activated AMPK, which further downregulated NFκB and led to a decrease in inflammatory markers such as cyclooxygenase-2, inducible nitric oxide synthase, IL-1β, IL-6, and TNF-α [43]. Moreover, irisin can exert its anti-inflammatory activity by binding to the extracellular domain of the transmembrane integrin αVβ5, along with the upregulation of AMPK and UCP2, promoting protection of intercellular tight junctions and a reduction in vascular permeability in inflammation [43]. It has also been shown that irisin can inhibit NFκB, c-Jun N-terminal kinases, and ERK phosphorylation through suppression of MAPK. In addition, irisin may protect against oxidative stress by lowering the expression of antioxidative enzymes like superoxide dismutase, glutathione peroxidase, and catalase 9 and reducing the hydrogen peroxide release from macrophages [26]. Lastly, irisin has been shown to promote cell survival and prevent apoptosis through the AKT, ERK, p38 MAPK, and TLR4/MyD88 signaling pathways [44].

The precise mechanisms of action of irisin are still not fully understood and require further investigation. However, existing evidence shows that this novel myokine holds great potential for becoming an important part of future therapies and prevention for many diseases. The overview of molecular mechanisms related to irisin and its role in GI tract diseases is shown in Figure 1 below.

## 6. Irisin and Inflammatory Bowel Disease

Irisin has been suggested as a potential novel therapy for IBD, targeting both inflammatory processes in the gut and remote changes in the bone [45,46,47,48]. IBD is a heterogeneous group of disorders of inflammatory conditions of the colon and small intestine that display two major phenotypic forms: Crohn’s disease (CD) and ulcerative colitis (UC) [49]. Their etiology is still not fully understood; however, aberrations in the secretion of cytokines such as interferon-γ, IL-5, IL-1β, IL-6, and TNF-α are undoubtedly involved in the pathogenesis of IBD. These proinflammatory cytokines were recognized to activate NFκB and MAPK, resulting in proinflammatory effects responsible for intestinal tissue pathology, including mucosal barrier disfunction and bacterial dysbiosis, as well as extra-intestinal comorbidities such as inflammation-induced bone loss [45,46,48,49,50].

Cytokines released in IBD can promote further inflammation, bone resorption, and alter the metabolism of osteocytes by increasing receptor activators of the NFκB ligand/osteoprotegerin receptor (RANKL/OPG). RANKL causes osteoclasts to differentiate and mature, leading to bone loss. OPG counteracts this effect by binding to receptor activators of NFκB receptors. An imbalance in the RANKL/OPG system is thought to be one of the main causes of bone loss in IBD patients [51]. The Wnt/β-catenin pathway has also been recognized to play a crucial role in bone mass regulation in IBD. Wnt/β-catenin activation leads to increased expression of genes related to bone formation and can exert anti-inflammatory effects in the bone. Wnt signaling pathway dysregulation has been found in intestinal mucosal samples from CD and UC patients [51]. Similarly, as in adipose tissue, irisin has been shown to target the Wnt signaling pathway in bone marrow mesenchymal stem cells. However, in Chen et al.’s [52] study, irisin first promoted autophagy. Only after the inhibition of autophagy did irisin activate the Wnt/β-catenin pathway, leading to an enhancement in osteogenesis [52].

The anti-inflammatory properties of irisin are manifested by attenuating the inflammation in IBD. In an experimental rodent model of mild IBD, treatment with exogenous irisin decreased the inflammatory markers during IBD, ameliorating changes in the colon, gut lymphatic structure, and bone tissue by reducing TNF-α and protein expression of RANKL [45]. Furthermore, in a rat model of severe IBD, treatment with irisin improved colon inflammation and intestinal histopathology but did not improve bone density or mechanical properties [46]. Bilski et al. [48] provided evidence for irisin’s involvement in the acceleration of mucosal healing promoted by moderate exercise in rats. In the study, plasma levels of irisin were significantly reduced in animals with induced colitis that were fed a high-fat diet [48]. Forced moderate physical training elevated levels of circulating irisin and, at the same time, decreased expression and blood levels of proinflammatory cytokines such as IL-1β and TNF-α [48].

Taken together, current evidence suggests that irisin could exert therapeutic effects in IBD through ameliorating inflammation processes both locally and in distant tissues, for instance by protecting against concomitant bone loss in IBD. Additionally, it may promote healing processes of the intestinal mucosal tissue and a healthy gut microbiome profile. However, it should be noted that the research on this topic is still very limited and is mainly based on animal models of IBD. Furthermore, the role of irisin in IBD has not been studied in clinical settings; therefore, human studies are very much needed and expected.

## 7. Irisin and the Gut Microbiome

Recently, a potential link between irisin and the gut microbiota has been suggested. Huangfu et al. [53] investigated the therapeutic effects of intraperitoneally administered irisin in a mouse model of experimental colitis. They found that irisin lowered the degree of inflammation in mice with colitis by reversing alterations to the macroscopic score, histological score, number of CD64+ cells, and inflammatory cytokine alterations [53]. Interestingly, these effects in colitis mice were accompanied by alterations in the biodiversity of the intestinal microbiota, changing the intestinal flora composition between the irisin-treated and control groups to be similar [53]. A study conducted on FNDC5 knockout mice further emphasized the significant role of irisin in preserving a healthy gut microbiome profile [54]. The knockout of the FNDC5 gene leads to a poorer diversity of gut microbiota and changes in microbiota-related metabolites [54]. Moreover, irisin has been shown to ameliorate lipopolysaccharide-induced inflammation in macrophages by MAPK/ERK and PI3K/AKT signaling pathways linked to the pathogenesis of UC and UC-associated colon cancer [55]. Kwon et al. [56] concluded that the PGC-1α pathway is involved in the beneficial effects of probiotic treatment in mouse models of adiposity, glucose intolerance, and dyslipidemia. Lactobacillus plantarum-treated animals showed upregulated expression of irisin in skeletal muscle and an improved profile of gut microbiota. These authors suggested that irisin, among other factors, may contribute to the beneficial metabolic changes induced by probiotic therapy [56].

## 8. Irisin and Oral Health

The latest research indicates the importance of oral health in the pathogenesis of GI diseases. Poor oral hygiene, periodontitis, stomatitis, and dysbiosis have been linked to the development of diseases such as IBD and GI malignancies, including colorectal and esophageal cancer [57,58,59]. Proposed direct and indirect mechanisms behind this association include migration of activated T cells through the blood and lymphatic systems, local metabolism of carcinogens, oral-to-gut translocation of microorganisms, and generalized chronic inflammation caused by oral diseases [50,51,52].

Existing evidence suggests that recreational physical activity may have a protective effect on periodontal health [60,61]. Indeed, current in vitro studies reveal the involvement of irisin in differentiation, growth, and migration, as well as extracellular matrix formation in various dental cell lines [62,63,64,65,66,67,68,69,70,71,72]. It has been reported that the p38 MAPK and AKT signaling pathways are involved in irisin’s effects [62,65,72]. The results of in vitro studies suggest that irisin may play a key role in promoting alveolar development and regeneration processes. Additionally, promising results were obtained from three different studies in animal models of periodontitis and orthodontic tooth movement [66,70,73]. Irisin treatment effectively reduced alveolar bone loss, decreased cytokine levels, and suppressed oxidative stress [66,70,73]. To our best knowledge, only two reports of human studies on irisin’s role in oral health have been published to date [74,75]. These studies have determined the concentration of irisin levels in the saliva of patients with periodontitis and healthy individuals. The concentration of irisin was increased in the periodontitis groups compared to the control groups; thus, the final conclusion was that salivary irisin may be a useful marker of periodontitis [74,75].

Overall, the results of in vitro and in vivo experiments consistently demonstrate the therapeutic potential and beneficial role of irisin in dental health. However, further exploration of clinical evidence is necessary, as only two human studies involving a small number of patients have been conducted. Given the promising results from preclinical experiments, there is an urgent need for comprehensive human research on the role of irisin in oral health.

## 9. Irisin and Intestinal Injury

Intestinal ischemia/reperfusion (IR) injury is a serious clinical entity that may result in intestinal mucosal injury, whose major underlying mechanisms include the generation of reactive oxygen species (ROS), the release of inflammatory mediators, and the induction of apoptosis [76]. It has been demonstrated in mice that serum and intestinal tissue irisin levels are decreased during intestinal IR and that treatment with exogenous irisin can restore the function of the intestinal epithelial barrier [23]. Moreover, another study [76] showed that pretreatment with irisin can reduce intestinal oxidative stress and tissue levels of proinflammatory cytokines like TNF-α, IL-1β, and IL-6. Additionally, irisin pretreatment downregulated Bax and cleaved caspase-3 at the protein level and increased the amounts of B-cell lymphoma 2 protein, significantly reducing apoptosis in the intestine of mice subjected to IR [76]. Furthermore, both in vivo and in vitro results presented in that study [76] revealed that irisin significantly upregulated the nuclear factor erythroid 2–related factor 2 (Nrf2) protein. Meanwhile, Nrf2 siRNA treatment partially abrogated the protective effects of irisin pretreatment against IR-induced cellular damage and its inflammatory response, oxidative stress, and apoptosis in IEC-6 cells [76]. These findings indicated that irisin improved the IR-induced intestinal inflammatory response by reducing oxidative stress and inhibiting apoptosis mediated, at least in part, by activating the Nrf2 pathway [76]. In another study, the administration of irisin significantly mitigated intestinal damage, reduced apoptosis, and attenuated oxidative and endoplasmic reticulum stress in mice with acute pancreatitis [77].

In conclusion, these experimental findings indicate that irisin has a potent therapeutic potential for ameliorating injuries induced by IR in the gut. It might counteract the deleterious processes involved in IR, like inflammation, oxidative stress, and cell apoptosis, while at the same time promoting healthy epithelial barrier function. Table 1 presents an overview of studies on irisin’s role in GI tract diseases.

## 10. Irisin and Gastrointestinal Malignancies

Epidemiological studies have shown that physical activity can significantly reduce the risk of developing various types of cancer and may inhibit their progression, which is beneficial in cancer therapy [78]. Among many molecular mechanisms implicated in cancer development, the downregulation of p53 is thought to play an important role in the process of carcinogenesis. The activation of the PI3K/AKT pathway has been shown to delay p53-mediated apoptosis [79]. Inflammation can directly promote cancer growth and development by generating a wide array of inflammatory molecules and biomarkers, including NFκB, hypoxia-inducible factor-1α, STAT3, and pro-inflammatory molecules such as TNF-α, IL-1β, IL-6, and IL-23. Furthermore, oxidative stress may play a critical role in cancer initiation and progression and is associated with mechanisms involving an increase in DNA damage, cell proliferation, and genomic instability [79,80].

Irisin has been shown to target multiple pathways responsible for protecting various tissues against stressful stimuli and conditions [79]. For instance, irisin initiated the apoptosis of malignant breast cancer cells through the activation of caspase-3 and caspase-7. The PI3K and AKT signaling pathways have been implicated in cell cycle regulation, apoptosis, and malignant transformation [80]. Irisin effectively reversed epithelial-mesenchymal transition (EMT) activity and suppressed Snail signaling through the PI3K/AKT pathway in lung cancer cells [80]. Irisin has also been shown to inhibit oxidative stress through the Nrf2/heme oxygenase-1 signaling pathway in the RAW 264.7 macrophage cell line, providing protection against ROS-induced damage under inflammatory conditions [79,81]. It has been demonstrated that irisin can inhibit the secretion of inflammatory cytokines, such as TNF-α, IL-6, and NFκB, and limit the recruitment of inflammatory T-cells and lymphocytes [79]. Moreover, it was observed that irisin may inhibit cancerogenic processes in pancreatic, breast, prostate, and lung cancers [82]. The mechanisms of the anticancer properties of this peptide depend on the inhibition of proliferation, survival, and migration of cancer cells [82]. Moreover, irisin is considered a potential therapeutic agent against obesity, which is a well-known risk factor for many malignancies, including cancers of the esophagus, colon, and rectum; however, its role in GI tract cancers has not been fully explained [82,83,84,85,86].

Using immunohistochemistry, Aydin et al. [85] traced irisin in different tissues of human GI cancers, including human esophageal epidermoid carcinoma, esophageal adenocarcinoma, neuroendocrine esophageal carcinoma, gastric adenosquamous carcinoma, gastric neuroendocrine carcinoma, gastric signet ring cell carcinoma, colon adenocarcinoma, and mucinous colon adenocarcinoma [85]. They noted a significant increase in irisin production in these tissues compared to normal tissues and offered two explanations for their observations [85]. First, increased irisin in cancerous tissues might be a way for the body to limit cell division by inhibiting ATP. Second, irisin can promote the browning of the fat tissue, which increases heat production by UCP1 rather than ATP. Thus, they proposed that irisin may increase local hyperthermia and induce cancer cell death by coagulating their proteins and destroying blood vessels [85]. Wozniak et al. [87] confirmed the above-mentioned findings using immunofluorescence on CaCo, LoVo, and HT29 colon cancer cell lines in comparison to normal colon cells. Moreover, they suggested that irisin could be useful for differentiation between the initial and more advanced stages of colon cancer [87].

In a clinical study involving 76 colon cancer patients and 40 healthy controls, serum irisin levels were significantly decreased in the cancer group [88]. The investigators suggested that irisin can play a role as an immense protective factor for colon cancer and might be used as a new diagnostic indicator capable of distinguishing colon cancer patients from healthy individuals [88]. Similar results were obtained in more recent clinical studies, where serum irisin levels were also lower in cancer groups as compared to healthy controls [89,90]. However, no correlation between levels of irisin and oxidative stress parameters was found [90]. Interestingly, Uzun et al. reported increased levels of irisin in patients with early stages of colorectal adenocarcinoma and decreased levels in the group at the advanced stage of this tumorigenesis [91].

Irisin has also been linked to weight loss in cancer cachexia, especially in gastric cancer [92]. In a study by Altay et al. [92], FNDC5 expression in brown adipose tissue and blood irisin levels in mice were significantly increased in cancer groups compared to the control group. Recent studies have confirmed that irisin levels are elevated in gastric cancers [93,94]. It has been hypothesized that irisin may inhibit EMT and the PI3K/AKT/Snail signaling pathway, controlling the migration and invasiveness of cancer cells. Increased irisin levels during the development of gastric cancer could also be considered a physiological response to oxidative stress [93]. Moreover, the increased amount of irisin in tumors may be associated with the higher rate of glycolysis in tumor cells, similarly to muscle cells, where irisin appears to increase glucose uptake by activating p38 MAPK through AMPKα2 and glucose transporter type 4 translocation to the plasma membrane [94]. Molfino et al. [95] observed that the modulation of different markers of browning of subcutaneous adipose tissue in GI cancer and pancreatic cancer showed significant changes in UCP1 and PGC-1α, the latter being highly expressed in patients with cancer cachexia [95]. However, Moon and Mantzoros [84] found no effect of irisin on cell proliferation or the malignant potential of human and mouse colon and esophageal cancer cell lines [84].

In conclusion, the role of irisin in GI cancers should be elucidated. The most promising discoveries regarding the anticancer properties of irisin come from studies on malignancies other than GI tract cancers. However, current evidence for the role of irisin in digestive tract pathophysiology shows that it may become a useful clinical marker for GI cancers. Moreover, it has been suggested that irisin may exert different effects depending on the type of cancer [79,95].

The summary of the beneficial effects of irisin in the therapy of the pathophysiology of the GI tract, including its favorable effect on the microbiota and different cancers, is presented in Figure 2. Table 2 provides an overview of studies on irisin’s potential role in GI malignancies.

## 11. Current Limits and Perspectives

Taken together, current evidence suggests that irisin could exert therapeutic effects in the GI tract through ameliorating inflammation processes both locally and in distant tissues, for instance, by its protecting effect against concomitant bone loss in IBD. However, there are some discrepancies regarding the detection of irisin in biological fluids. So far, the majority of studies have involved irisin ELISA kits, but their usefulness raises concerns concerning the range of reliable values and detection limits. Nevertheless, the evident protective and healing effects of irisin, especially those observed in the intestinal mucosal tissue, and its beneficial influence on restoring a healthy gut microbiome profile observed in animal models of IBD encourage further basic and clinical research. Furthermore, more clinical studies on the role of irisin and its receptors are expected in human upper- and lower-GI tract disorders, including peptic ulcer disease, IR lesions, and IBD. The prosperous role of irisin in maintaining a healthy intestinal microbiome profile also requires further investigation. Considering the importance of the gut microbiome to overall health, research indicating the possible use of irisin as an adjuvant to probiotic therapy nowadays may bring benefits that extend beyond the digestive system. Therefore, there is an unmet need for experimental and clinical research in GI tract disorders, including various cancers, to further investigate the role of irisin and this peptide’s potential therapeutic significance.

## 12. Conclusions

The protective potential of irisin in the GI tract has not yet been widely investigated, and further high-quality in vivo and in vitro research, as well as clinical trials, are needed to determine whether irisin can be used for the prevention and treatment of various GI disorders. Moreover, the role of irisin in various digestive tract cancers seems complex and is still not fully understood. However, the recent accumulation of evidence highlights the prominent beneficial anti-inflammatory and anticancer properties of irisin. These findings suggest that irisin may serve as a protective factor with therapeutic potential against certain diseases and cancers affecting the digestive tract.

## Figures and Tables

**Figure 1 antioxidants-13-00413-f001:**
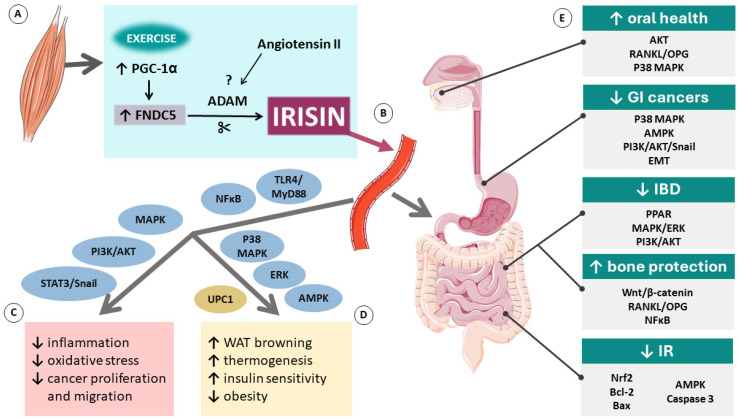
Molecular mechanisms related to irisin: its production in the muscle (**A**), release to the bloodstream (**B**), generalized effects in the body (**C**), fat tissue (**D**), and gastrointestinal tract (**E**). Abbreviations: ↑ indicates an increase; ↓ indicates a decrease; ADAM, disintegrin and metalloproteinase; AKT, protein kinase B; AMPK, AMP-activated protein kinase; Bcl-2, B-cell lymphoma 2; EMT, epithelial-mesenchymal transition; ERK, extracellular signal-regulated kinase; FNDC5, fibronectin type III domain containing 5; GI, gastrointestinal; IBD, inflammatory bowel disease; IR, ischemia/reperfusion; MAPK, mitogen-activated protein kinase pathway; NFκB, nuclear factor-κ light chain enhancer of activated B cells; Nrf2, nuclear factor erythroid 2–related factor 2; PGC1-α, co-activator PPAR-γ co-activator-1 α; PI3K, phosphatidylinositol 3 kinase; PPAR, peroxisome proliferator-activated receptor; RANKL/OPG, receptor activator of NFκB ligand/osteoprotegerin receptor; STAT3, signal transducer and activator of transcription 3; TLR4/MyD88, Toll-like receptor 4/myeloid differentiation primary response 88; UCP1, mitochondrial uncoupling protein 1. Parts of the figure were drawn using pictures from Servier Medical Art. Servier Medical Art by Servier is licensed under a Creative Commons Attribution 3.0 Unported License (https://creativecommons.org/licenses/by/3.0/).

**Figure 2 antioxidants-13-00413-f002:**
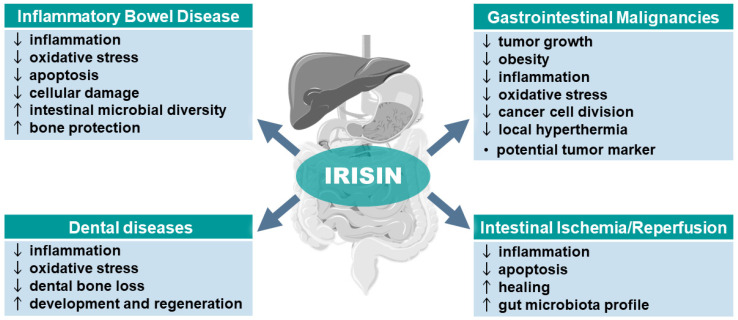
Potential role of irisin in gastrointestinal tract diseases. Abbreviations: ↑ indicates an increase or improvement; ↓ indicates a decrease or worsening. Parts of the figure were drawn using pictures from Servier Medical Art. Servier Medical Art by Servier is licensed under a Creative Commons Attribution 3.0 Unported License (https://creativecommons.org/licenses/by/3.0/).

**Table 1 antioxidants-13-00413-t001:** Studies on irisin’s role in gastrointestinal tract diseases.

Disease	Organism/Model/Cell Line	Methods/Intervention	Irisin Detection Method	Findings/Effects of Irisin	Signaling Pathways	Ref.
IBD	Rats with TNBS-induced colitis	HFD, LFD, ND, forced exercise	Plasma irisin, ELISA	↓ irisin in the HFD colitis group ↑ irisin in exercising groups	N/A	[48]
IBD	Mice with TNBS-induced colitis	HFD, ND, voluntary exercise	Plasma irisin, detection method not specified	↓ irisin in the HFD sedentary group ↓ FNDC5 gene expression in WAT of colitis mice	N/A	[49]
IBD	Rats with TNBS-induced colitis	Intraperitoneal injection of irisin (18 ng/mL)	N/A	↓ inflammatory markers ↑ colonic lymphatic structure ↑ bone formation	RANKL/OPG	[45]
IBD	Mice with DSS-induced colitis	Intraperitoneal injection of irisin (0.0075 µg/g)	N/A	↓ macroscopic and histopathological scores ↓ colonic CD64+ cells ↓ plasma IL-12 and IL-23; ↑ gut microbiota	N/A	[53]
UC, LPS-induced inflammation	LPS-induced macrophage (RAW 264.7) cell line	Incubation with irisin (400 ng/mL) and/or LPS in vitro	N/A	↓ inflammation ↓ cytotoxicity and apoptosis ↓ IL-12 and IL-23	MAPK, ERK, PI3K/AKT, PPAR	[55]
Gut dysbiosis	FNDC5-knocked out mice	Fecal microbiota assessment	N/A	↑ dysbiosis of the gut microbiota	N/A	[54]
Periodontitis	Mouse cementoblast (OCCM-30) cell line	Incubation with irisin (0–200 ng/mL)	N/A	↑ differentiation ↑ proliferation ↑ mineralization	p38 MAPK	[72]
Periodontitis	Human osteoblast cell lines	Incubation with irisin (10 and 100 ng/mL)	N/A	↑ proliferation ↑ extracellular matrix formation ↑ migration	N/A	[63]
Dental	Human dental pulp cells	Incubation with irisin (5, 10, 20, 40 µM)	N/A	↑ differentiation ↑ mineralization ↑ angiogenesis	MAPK, AKT	[65]
Periodontitis	Human patients	Periodontitis assessment	Salivary irisin, ELISA	↑ salivary irisin in periodontitis	N/A	[74]
Periodontitis	Human patients	Periodontitis assessment	Salivary irisin, ELISA	↑ salivary irisin in periodontitis	N/A	[75]
IR	IR-induced mice, HR simulated rat intestinal epithelial (IEC-6) cell line	Intravenous injection of irisin (10 and 100 ng/g) in vivo; irisin pretreatment (1, 10, 100 ng/mL) in vitro.	N/A	↓ oxidative stress ↓ apoptosis ↓ inflammatory markers	Nrf2, Bcl-2/Bax, Caspase-3	[76]
IR	IR-induced mice, HR simulated human colon carcinoma (Caco-2) cell line	Intravenous injection of irisin (250 μg/kg) in vivo; irisin treatment after HR (10 nmol/L) in vitro	Western blot	↑ gut barrier function ↓ oxidative stress	αVβ5-AMPK-UCP2	[23]
IR, AP	AP-induced mice	Intraperitoneal injection of irisin (50 and 250 μg/kg)	N/A	↓ intestinal damage ↓ apoptosis ↓ oxidative stress	N/A	[77]

Abbreviations: ↑, increase or improvement; ↓, decrease or worsening; N/A, not applicable; AKT, protein kinase B; AMPK, cyclic adenosine monophosphate-activated protein kinase; AP, acute pancreatitis; Bcl-2, B-cell lymphoma 2; DSS, dextran sodium sulfate; ELISA, enzyme-linked immunosorbent assay; ERK, extracellular signal-regulated kinase; FNDC5, fibronectin type III domain containing 5; HFD, high-fat diet; HR, hypoxia/reoxygenation; IBD, inflammatory bowel disease; IL-, interleukin-; IR, ischemia/reperfusion; LFD, low-fat diet; LPS, lipopolysaccharide; MAPK, mitogen-activated protein kinase pathway; ND, normal diet; Nrf2, nuclear factor erythroid 2–related factor 2; PI3K, phosphatidylinositol 3-kinase; PPAR, peroxisome proliferator-activated receptor; RANKL/OPG, receptor activator of NFκB ligand/osteoprotegerin receptor; TNBS, 2,4,6-trinitrobenzenesulfonic acid; UC, ulcerative colitis; UCP2, mitochondrial uncoupling protein 2; WAT, white adipose tissue.

**Table 2 antioxidants-13-00413-t002:** Studies on irisin’s role in gastrointestinal tract malignancies.

Disease	Organism/Model/Cell Line	Methods/Intervention	Irisin Detection Method	Findings/Effects of Irisin	Ref.
Colon cancer, esophageal cancer	Human and mouse colon (HT29, MCA38) and esophageal (OE13, OE33) cell lines	Incubation with irisin (5, 10, 50, and 100 nM)	N/A	No effects on cell proliferation or malignant potential	[84]
Various GI cancers	Human patients	Collection and histopathological evaluation of cancer tissues	Immunohistochemical staining	↓ irisin in GI cancer tissues	[85]
Gastric cancer	Mice	Carcinogen administration	Serum irisin, ELISA, Real-time PCR	No irisin/FNDC5 gene expression in gastric tissue, ↑ irisin/FNDC5 gene expression in fat tissues in the cancer group	[92]
Gastric cancer	Human patients	Blood sample collection and analysis	Serum irisin, ELISA	↑ serum irisin in cancer patients	[93]
Gastric and colorectal cancer	Human patients	Blood and tissue sample collection and analysis	Multiplex technology with the Magpix instrument	↑ irisin in tumor tissue of patients with cancer cachexia	[94]
Gastric and colorectal cancer	Human patients	Blood sample collection and analysis	Serum irisin, ELISA	No correlation between irisin and oxidative stress, ↓ serum irisin in cancer patients	[90]
Colorectal cancer	Human patients	Blood and tissue sample collection and analysis	Serum irisin, ELISA	↓ serum irisin in cancer patients	[88]
Colorectal Cancer	Human patients, human intestinal (CCD-18Co) and intestinal cancer (CaCo-2, LoVo, HT-29) cell lines	Histopathological evaluation of colorectal samples, in vitro cell culture	Immunohistochemical staining, immunofluorescence, western blot	↑ irisin in cancer cells ↑ irisin in initial colon cancer ↓ irisin in more advanced stages	[87]
Colorectal cancer	Human patients	Tissue sample collection and analysis	Immunohistochemical staining	↑ irisin in initial cancer, ↓ irisin in more advanced stages	[91]
Colorectal cancer	Human patients	Blood sample collection and analysis	Serum irisin, ELISA	↓ serum irisin in cancer patients	[89]

Abbreviations: ↑, increase or improvement; ↓, decrease or worsening; N/A, not applicable; ELISA, enzyme-linked immunosorbent assay; FNDC5, fibronectin type III domain containing 5; GI, gastrointestinal; PCR, polymerase chain reaction.

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
