# Peer review of "The Role of the Myokine Irisin in the Protection and Carcinogenesis of the Gastrointestinal Tract"

_antioxidants, 2024, doi:10.3390/antiox13040413_

Round 1
Reviewer 1 Report
The review presented by Pinkas and Brzozowski is interesting and well written. It reports the state of the art on the possible role of myokine irisin in the affections of the grastointestinal tract. In the first part was described the myokine and its possible mechanisms of action and detection methods; in the second, the evidence on its effects in the oral-gastro-intestinal disorders twere reported.
To make the review more complete it would be interesting to describe in a separate paragraph the interaction of irisin with the intestinal microbiota.
Figures and tables are very clear.
Improving the figure resolution and word colors.
Author Response
Point by point response to Reviewer #1
We would like to express our gratitude to Reviewers for their meaningful and helpful critical comments on our manuscript in original version. All critical comments were carefully considered, and the paper was revised in accordance with all Reviewers comments.
Reviewer #1:
Thank you for your positive outcome on our review dedicated to the role of irisin in gastrointestinal tract. We revised our work based on your kind critical comments as follows:
Rev: To make the review more complete it would be interesting to describe in a separate paragraph the interaction of irisin with the intestinal microbiota.
AU: The research on the interaction of irisin with the intestinal microbiota is still very limited, however, we share similar opinion with Reviewer that this issue is of great interest nowadays. Therefore, we have moved the last paragraph of the chapter number 6 entitled “Irisin and inflammatory bowel disease” to a new section titled “Irisin and gut microbiome” according to suggestion proposed by this Reviewer.
Rev: Improving the figure resolution and word colors.
AU: Thank you for useful comment bringing Figure colors and their resolution into our attention. We edited the Figures to make them with straightforward text and greater visibility. The original file containing Figures will be provided along with the revised review paper for best resolution of the images.
Reviewer 2 Report
The manuscript antioxidants-2905183 reviews the role of myosin, a myokine produced after physical activity, as a potential beneficial molecule for the gastrointestinal tract. Overall, the manuscript is well written, and figures and tables are useful to summarise the main concepts. However, I believe that minor improvements might be needed before publication.
I would like the authors to address and discuss the following points.
* It has been shown that some individuals are not responder in terms of irisin released after physical activity and that further research should address the actual effect of exercise on irisin concentrations and the physiological consequences of these increases (https://www.ncbi.nlm.nih.gov/pmc/articles/PMC6286984/). Can the authors please elaborate more on this?
* In the past, some authors speculated against a beneficial effect of irisin in humans supported by inconsistencies between animal and human data (https://www.ncbi.nlm.nih.gov/pmc/articles/PMC3770677/). This discrepancy is still highlighted also in the Uniprot database https://www.uniprot.org/uniprotkb/Q8NAU1/entry#function. Can the authors please elaborate more on this?
* Can the authors comment on the type of exercise that is supposed to be better in raising irisin levels?
* The information contained in paragraph 4 (technical issues and limits for detecting irisin in biological fluids) and comments about the need of further studies found at the end of paragraphs 6 (lines 251-254), 7 (lines 278-283), 8 (lines 308-309), and 9 (lines394-401) can be moved into a separate, final paragraph discussing current limits and future perspectives.
Author Response
Point by point response to Reviewer #2
We appreciate your insightful comments to our review on the role of irisin in health and disease of digestive system. All critical comments were carefully considered, and paper was revised in accordance with this Reviewer comments and suggestions.
Rev: It has been shown that some individuals are not responder in terms of irisin released after physical activity and that further research should address the actual effect of exercise on irisin concentrations and the physiological consequences of these increases (https://www.ncbi.nlm.nih.gov/pmc/articles/PMC6286984/). Can the authors please elaborate more on this?
AU: We agree that the lack of response by satisfactory levels of irisin despite exercise in some individuals is an important aspect that should be addressed in our review. We have revised the text elaborating on this topic by adding some results of the research work reflecting not satisfactory levels of irisin in exercising individuals.
Rev: In the past, some authors speculated against a beneficial effect of irisin in humans supported by inconsistencies between animal and human data (https://www.ncbi.nlm.nih.gov/pmc/articles/PMC3770677/). This discrepancy is still highlighted also in the Uniprot database https://www.uniprot.org/uniprotkb/Q8NAU1/entry#function. Can the authors please elaborate more on this?
AU: We absolutely agree that, in general, the data accumulated in animals cannot be always translated to humans, and i.e. irisin’s discovery was met with criticism and controversies in the first years following this peptide discovery. The article by Raschke et al. mentioned in your review provided valuable concerns regarding the start codon of the FNDC5 gene in humans and the ability of irisin to induce browning of human adipocytes. On the other hand, for instance, recent studies clearly supported the role of irisin in the mechanism of browning of human adipose tissue (e.g. https://doi.org/10.1152/ajpendo.00094.2016). Thanking you for raising this point, we have addressed these concerns underlined by work of Raschke et al. in the revised version of this manuscript (please see, lines 131-135).
Rev: Can the authors comment on the type of exercise that is supposed to be better in raising irisin levels?
AU: The most effective training protocol for increasing circulating levels of irisin has still not been well established. Based on our own experience, the voluntary moderate exercise is the best choice to improve the gastrointestinal functions, however, our review refers to current evidence on irisin response to various types of exercise in different populations. Our intention was to select the most relevant recent original findings and meta-analyses related to this topic.
Rev: The information contained in paragraph 4 (technical issues and limits for detecting irisin in biological fluids) and comments about the need of further studies found at the end of paragraphs 6 (lines 251-254), 7 (lines 278-283), 8 (lines 308-309), and 9 (lines394-401) can be moved into a separate, final paragraph discussing current limits and future perspectives.
AU: We are grateful to Reviewer for her/his suggestion to re-arrange some paragraphs and to briefly discuss on limits and perspectives of irisin science. Suggested section were moved into a separate paragraph entitled “Current Limits and Future Perspectives” which provides some difficulties and limits in irisin research (please see, subparagraph # 11). Once again, we would like to acknowledge this Reviewer thoughtful comments and suggestions to improve our manuscript in the final version.